# Oral Health in 12- and 15-Year-Old Children in Serbia: A National Pathfinder Study

**DOI:** 10.3390/ijerph191912269

**Published:** 2022-09-27

**Authors:** Tamara Peric, Guglielmo Campus, Evgenija Markovic, Bojan Petrovic, Ivan Soldatovic, Ana Vukovic, Biljana Kilibarda, Jelena Vulovic, Jovan Markovic, Dejan Markovic

**Affiliations:** 1Clinic for Pediatric and Preventive Dentistry, School of Dental Medicine, University of Belgrade, 11000 Belgrade, Serbia; 2Department of Restorative, Pediatric and Preventive Dentistry, Dental Clinic, University of Bern, 3010 Bern, Switzerland; 3Clinic of Orthodontics, School of Dental Medicine, University of Belgrade, 11000 Belgrade, Serbia; 4Department of Pediatric and Preventive Dentistry, Dentistry Clinic of Vojvodina, Faculty of Medicine, University of Novi Sad, 21000 Novi Sad, Serbia; 5Department of Statistics and Bioinformatics, School of Medicine, University of Belgrade, 11000 Belgrade, Serbia; 6Institute of Public Health of Serbia “Dr. Milan Jovanović Batut”, 11000 Belgrade, Serbia; 7Faculty of Medicine, University of Pristina-Kosovska Mitrovica, 38220 Kosovska Mitrovica, Serbia

**Keywords:** oral health, schoolchildren, Serbia

## Abstract

The aim of the paper is to present the oral health profile of 12- and 15-year-old schoolchildren in Serbia. Basic Methods for Oral Health Surveys of the WHO were implemented to record: Decayed, Missing, and Filled Teeth/Surfaces Index (DMFT/DMFS), gingival bleeding, enamel fluorosis and other structural anomalies, dental erosion, dental trauma, and oral mucosal lesions. In addition, Silness and Löe plaque index and orthodontic status were assessed. A total of 36% of 12-year-olds and 22% of 15-year-olds in Serbia were caries-free. The mean DMFT was 2.32 ± 2.69 for 12-year-olds and 4.09 ± 3.81 for 15-year-olds. DMFT was made up largely by the decayed component. Gingival bleeding was present in 26% of examined 12-year-old and 18% of 15-year-old children. Dental plaque was observed in 63% of both 12- and 15-year-olds. Fluorosis, structural anomalies, dental erosion, dental trauma, and oral mucosal lesion were rarely detected. Low prevalence of malocclusions was found. Oral disease is still a common public health problem among schoolchildren in Serbia. A significant increase in the prevalence of caries disease between 12- and 15-year-old groups implies that preventive care for adolescents requires special attention. Corrective actions and reforms to the current school-based oral health prevention program are needed to further improve oral health in Serbian children.

## 1. Introduction

The Republic of Serbia is a European country situated in the West-Central Balkans. According to the latest census from 2011 [1], Serbia (without the Kosovo and Metohija region) has a population of 7,186,862 people. It consists of four administrative regions: (1) the capital—the City of Belgrade (1,659,440 inhabitants), (2) the northern region—Vojvodina (1,931,809 inhabitants), (3) Central and Western Serbia (2,031,697 inhabitants), and (4) Southern and Eastern Serbia (1,563,916 inhabitants).

During previous decades, several national oral health surveys of representative samples were conducted in the Republic of Serbia. During the socialist era in the second half of the 20th century, Serbia had a government-controlled health care system directed to provide equal distribution of health care. All costs, including dental services, were covered by the state [2]. The first national epidemiological study in former SFR Yugoslavia was conducted in 1986 [3]. The study revealed that more than 90% of children in all age groups had caries, and the mean DMFT was 6.1 for 12-year-olds. Subsequently, the first oral health preventive program was established. However, in the 1990s, Serbia was burdened by the financial consequences of the Yugoslav wars and economic sanctions, which had a significant negative impact on implementation and dissemination of the preventive program in the community. Political changes in the beginning of the 21st century brought a restructuring of the health care system. The impact of oral diseases on general health and well-being has not been addressed adequately in Serbian society, even amongst medical professionals. In addition, economic reasons significantly influenced health care policies, and reduced the extent of dental services funded by the public health care system. Nowadays, preventive and restorative oral health care services are funded by the government in public health care institutions for children and adolescents up to 18 years [4]. 

The latest report on oral health in schoolchildren and adolescents in Serbia was published in 2009 [5]. Since then, new trends in preventive dentistry and cariology have been established, but no science-based modifications and improvements were introduced into the program. A government initiative for updating preventive programs was announced in 2019, intended to develop a modernized program for the prevention of oral diseases in children and adolescents in Serbia. The first step in making a restructured preventive program was to collect epidemiological data in a national oral health survey. The aim of this paper is to present the oral health profile of 12- and 15-year-old schoolchildren in Serbia.

## 2. Materials and Methods

The present study was part of a national pathfinder survey that aimed to evaluate the oral health status in children and adolescents in Serbia. The survey was designed according to the WHO methodology for oral health surveys [6], approved by the Ethics Committee at the University of Belgrade School of Dental Medicine (document 36/10), and supported by the Ministry of Health of the Republic of Serbia (document 500-01-49-3/2019-07).

According to the WHO [6], in populations where the disease level is known to be high, namely the percentage of caries-free 12-year-olds is less than 20%, the proposed number of subjects in each index age group for each sampling site is approximately 50 subjects. The results of the last epidemiological study in Serbia from 2009 showed that the level of caries-free 12-year-olds was 18% [5]. Therefore, the proposed sample design for oral health survey in Serbia was 50 children of each age group to be examined at the following 14 sampling sites: 4 sites in the capital city (2 urban, 2 periurban), 6 sites in 3 large towns located in 3 administrative regions (1 urban, 1 periurban) and 1 site in 4 rural areas. Such a sample design provides a minimum of 700 participants in each age group.

Prior to the survey, selected schools and health centers were reached for the approval, and written informed institutional consents were obtained. Dental practitioners from 33 pediatric dental services in Serbia were contacted to participate in the national survey, and enter the calibration process. For the purpose of the study, the calibration of examiners was performed at two levels. At the first level, calibration of three principal investigators was completed. Afterwards, experienced pediatric dentists—examiners, completed a two-day training course in oral examination, and calibration of all investigators was performed. Each of three principal investigators supervised and calibrated one group of 11 examiners from one geographical location at one of 33 sampling sites. The results of examiners’ consistency after examinations of permanent teeth are presented in Table 1 and Table 2.

Eventually, 24 school dental services were selected to participate in the study, and schoolchildren were evaluated at the following sites: 3 urban and 2 periurban sites in the capital city of Belgrade, 3 urban, 1 periurban, and 2 rural sites in Vojvodina, 3 urban, 1 periurban, and 3 rural sites in Central and Western Serbia, and 3 urban, 1 periurban, and 2 rural sites in Southern and Eastern Serbia.

Informed parent consent was obtained in writing prior to the child’s participation in the study, and informed assent was obtained from the child.

Oral examinations were undertaken in school dental ambulances using a dental mirror and probe between March and July 2019. The WHO Oral Health Assessment Form for Children was translated to Serbian and used in order to record the following: dentition status (Decayed, Missing, and Filed Teeth (DMFT index) and Decayed, Missing, and Filed Teeth Surfaces (DMFS index)), periodontal status (gingival bleeding), presence of enamel fluorosis and other structural anomalies, dental erosion, dental trauma, and oral mucosal lesions [6]. In addition to cavitated caries lesions (ICDAS 3–6 [7]), visual changes in enamel (ICDAS 1–2 [7]) were recorded. Children exhibiting high scores for one or more of the following conditions: DMFT ≥ 5, D ≥ 3, M ≥ 1, and PUFA index [8] ≥ 1, were grouped as those with severe caries [9]. Furthermore, Silness and Löe plaque index (PI) [10] and orthodontic status were assessed.

Depending on data type and distribution, results are presented as count (%), means ± standard deviation, or median. Groups were compared using a non-parametric chi-square, Mann–Whitney U test, or Kruskal–Wallis test. *p* values lower than 0.05 were considered significant. The data were analyzed using the SPSS 20.0 (IBM Corp. Released 2011. IBM SPSS Statistics for Windows, Version 20.0. Armonk, NY, USA: IBM Corp.) and R 3.4.2. (R Core Team (2017). R: A language and environment for statistical computing. R Foundation for Statistical Computing, Vienna, Austria).

## 3. Results

The present survey included 1200 12-year-olds and 1220 15-year-olds of both genders.

Thirty-six percent of 12-year-olds and 22% of 15-year-olds in Serbia were caries-free. If only obvious dental decay experience was considered, 47% of 12-year-olds and 28% of 15-year-olds in Serbia were caries-free. Of children diagnosed with caries, 44% of 12-year-olds and 61% of 15-year-olds were diagnosed with severe form.

The mean DMFT was 2.32 ± 2.69 (median 2.00) for 12-year-olds and 4.09 ± 3.81 (median 3.00) for 15-year-olds. For 12-year-olds, mean number of decayed teeth was 1.34 ± 2.12, and mean number of restored teeth was 0.91 ± 1.54. For 15-year-olds, mean number of decayed teeth was 2.08 ± 2.89, and mean number of restored teeth was 1.87±2.49. The mean DMFS was 3.16 ± 4.48 (median 2.00) for 12-year-olds and 5.83 ± 6.42 (median 4.00) for 15-year-olds. The share of surfaces with initial caries was 45% for 12-year-olds and 39% for 15-year-olds.

Dental status according to gender, area, administrative region, and parental employment status is presented in Table 3 for 12-year-olds and Table 4 for 15-year-old Serbian schoolchildren. Twelve-year-old girls had significantly higher DMFT/DMFS scores than boys (*p* < 0.001, Mann–Whitney test, Table 3). Children who had only one working parent had higher scores than children with both parents working (*p* < 0.05, Kruskal–Wallis test; Table 3 and Table 4). Children living in urban areas had lower scores compared to suburban and rural sites. Significantly lower DMFT/DMFS values were found in children residing in the capital city of Belgrade (*p* < 0.001, Kruskal–Wallis test; Table 3 and Table 4).

DMFT/DMFS was made up largely by the decayed component (D 58%, M 3%, F 39%; DS 57%, FS 33% for 12-year-olds, and D 51%, M 4%, F 45%; DS 47%, FS 40% for 15-year-olds). There was a significant difference in the distribution of D and F components for both 12- and 15-year-olds between the regions (*p* < 0.001, Kruskal–Wallis test, Figure 1). In the group of 15-year-olds, children from rural locations had more untreated caries present than children from urban and suburban locations (Figure 2). 

Overall, 15% of 12-year-old children and 16% of 15-year-olds had fissure sealants present. The occurrence of fissure sealing was significantly lower in Eastern and Southern Serbia (*p* < 0.001, chi-square test). There were no differences in DMFT values between 12-year-old children who had fissure sealants present (2.40 ± 2.78, median 2.00) and those who had not (1.85 ± 2.09, median 1.00) (*p* > 0.05, Mann–Whitney test). However, DMFT scores were significantly lower in 15-year-olds with sealants (3.15 ± 3.21 (median 2.00)), in comparison to children without sealants (4.27 ± 3.88 (median 4.00)) (*p* < 0.05, Mann–Whitney test).

Gingival bleeding was present in 26% of examined 12-year-old and 18% of 15-year-old children. Dental plaque was observed in 63% of both 12-year-olds (PI = 0.71 ± 0.78) and 15-year-olds (PI = 0.61 ± 0.72) (Table 5).

Fluorosis, structural anomalies, dental erosion, and oral mucosal lesion were rarely detected (<1%), while dental trauma was noted in 2% of children. Orthodontic status is shown in Table 6.

## 4. Discussion

This was the first study in Serbia, under the umbrella of the WHO, to provide a better insight into all aspects of oral health, including oral hygiene and orthodontic status in 12- and 15-year-old schoolchildren. The importance of this study in the country that has had many challenges in past decades is enormous. Even in high-income countries with great funding for national research projects, pathfinder surveys are rare [11].

The investigation of oral health trends in young population is a powerful instrument in monitoring efficacy of preventive measures. In comparison to the findings of previous epidemiological studies, there was a noticeable reduction in carious disease in schoolchildren and adolescents in Serbia. According to the latest report on oral health published in 2009 [5], 18% of 12-year-olds and 7.8% of 15-year-olds were caries-free, and the mean DMFT was 3.3 and 5.5, respectively. Findings of the present survey show that 36% of 12-year-olds and 22% of 15-year-olds in Serbia are caries-free (47% of 12-year-olds and 28% of 15-year-olds, if only obvious decay is considered). The number of caries-free children in the last 10 years has doubled, which is a good sign of progress in the prevention of oral diseases in Serbia. The positive trend could be attributed to easier access to dental care, better education, and improved socio-economic status in the country. Significant improvements of oral health awareness, knowledge, and attitudes were also reported in previous studies during the last decades [12].

According to the WHO caries severity criteria [6], Serbian schoolchildren currently belong to the low-risk group (mean DMFT in 12-year-olds: 1.2–2.6), compared to the moderate-risk group (mean DMFT in 12-year-olds: 2.7–4.4) in the past. Conversely, the increase in mean DMFT/DMFS between 12- and 15-year-old groups is a worrying trend. Adolescence is a period of numerous behavioral shifts [13], including change in dietary habits, negligence of oral hygiene, and possible adoption of nicotine, alcohol, and drugs addiction. Although adolescents may be perceived as responsible and capable of taking care of their oral health, it does not seem to be so. It has been shown that sugar consumption was higher during adolescence [14], and motivation for brushing teeth and using proper brushing technique may vary greatly [15]. Thus, prevention and treatment programs for adolescents should be emphasized and designed with special care.

Nowadays, initial changes in enamel are considered a disease requiring therapeutic response. This approach reflects changes in diagnosis and clinical practice towards preventive and minimum intervention approach to caries management [9]. Therefore, in the present study, besides identifying obvious cavitated lesions, non-cavitated caries lesions were also noted. Of all recorded decayed tooth surfaces, 45% in 12-year-olds, and 39% in 15-year-olds were initial caries lesions. These data clearly show that more rigorous preventive measures could further improve the caries status in Serbian children. One of the most effective prophylactic measures is fissure sealing. The present study revealed that the presence of fissure sealants was related to lower DMFT values. This finding is in accordance with the results of previous studies, which identified fissure sealing as a strong predictor for good oral health [16,17,18].

Significant share of severe caries has been noted among children with caries. Perhaps even more worrying is a larger proportion of untreated teeth with decay rather than restored teeth. This finding is in line with the conclusions of Kassebaum et al. [19], who showed that the burden of untreated caries was higher in Eastern and Central Europe compared to Western Europe. The mean DMFT value in Western European countries was lower, probably due to well-organized public oral health services for children, and high socio-economic status of the population, which enable widespread use of fluoridated and other oral hygiene products [20,21]. On the contrary, results of recent epidemiological studies from other parts of the world, such as Eastern European countries, Latin America, Africa, and Asia are not very encouraging, showing no improvement in caries experience [22,23,24,25,26,27,28,29,30,31,32,33,34]. When it comes to the epidemiological studies performed in the Western Balkan Region, i.e., countries constituted after the Yugoslavia breakup, data on oral health are scarce. To the best of our knowledge, the only oral health survey at the national level was conducted in Bosnia and Herzegovina in 2004 [35]. Similar to the findings of the present study, the D component was the major part of the DMFT index in 12-year-old Bosnian children. That was also the finding from the pathfinder survey in the neighboring Romania, especially in children from rural areas [36].

The present survey revealed that the capital city of Serbia, Belgrade had significantly more caries-free children and lower DMFT/DMFS values compared to other regions. The distribution of D and F components was balanced in all regions, except for Eastern and Southern Serbia where the prevalence of untreated and severe caries was significantly higher. In addition, the share of decayed and extracted teeth increased from urban to rural locations. The capital city of Belgrade is the most developed region in the country with the highest proportion of highly educated people [1]. On the contrary, Southern Serbia is less developed region, with low socio-economic status, which might affect oral health, provision of dental care, and implementation of preventive measures. The same may be the reason for differences in the oral health of children residing in urban, suburban, and rural sites. It has been emphasized that oral health knowledge, attitudes, and practices differ between urban and rural populations, as well as between privileged and underprivileged people [37]. Additional problems may be a limited access to dental health care in rural areas, and less opportunity for adequate provision of preventive measures and dental services [13,38,39,40,41]. Another reason for the high prevalence of severe and untreated caries might be the distribution of working dental professionals in the public dental system. According to 2019 annual reports, Serbia has 1,276,654 children aged 0–18 years [42], and the number of pediatric dentists in public health institutions is 322 [43]. That means that each pediatric dentist provides dental care for an average of 3965 children, which is not in accordance with good practice recommendations (one pediatric dentist per maximum 1500 children). 

As previously shown [13], children’s oral health was influenced by parental employment status. Higher caries incidence, higher DMFT/DMFS scores, and worse oral hygiene were observed in children with only one working parent compared to children with both working parents. Lower socio-economic status might determine poor oral health [44,45], probably due to the unaffordable oral hygiene products and high-sugar diets. Interestingly, the findings of the present study showed that the level of oral health in children whose parents were both unemployed was similar to children with both working parents.

The present study showed that gingival bleeding was observed in every fourth 12-year-old, and every fifth 15-year-old in Serbia, which was in accordance with the findings of previous studies in Europe [20]. The incidence of gingivitis varies globally [46] due to differences in oral habits and availability of oral hygiene products. In addition, methodological differences between the studies might influence the results. The community periodontal index (CPI) proposed by the WHO [6] has been widely used because it is simple and fast to use, but it has its limitations [47]. The CPI only shows whether gingivitis exists, but does not show the severity of the disease. Findings of the present study showed that oral hygiene in Serbian children was not adequate, as noticeable dental plaque was observed in two-thirds of schoolchildren. Similar findings [44] show the need for consistency in public health work to raise children’s awareness of the importance of maintaining oral hygiene.

Non-carious dental lesions, such as developmental defects and erosive tooth wear, have been the subject of growing interest [48]. Numerous studies investigated the prevalence of abnormalities in tooth size, shape, number, position, and structure in various countries and different ethnic groups. However, those studies showed methodological variations in sampling and choice of diagnostic methods, which led to inconsistent and incomparable results [49]. A recent study [50] reported higher prevalence of anomalies in tooth position and shape compared to the anomalies of the tooth structure. The overall prevalence of structural abnormalities was 0.2%, which corresponds to the findings of the present study.

Low incidences of fluorosis were found in previous [3] and present national surveys in Serbia in comparison to similar studies in Europe [51] and America [48]. Fluoride concentration in soil and water varies among the regions in Serbia. In some areas, the average level of fluoride in wells is more than 10 times higher than the recommended value. The first law on community water fluoridation in Serbia was created in 1971. Water fluoridation was implemented in several towns (Pancevo, Uzice, Krusevac, Cacak), but it was discontinued during the 1990s due to a lack of funding [2]. Nowadays, the majority of inhabitants in Serbia are drinking tap water, which is not fluoridated.

Erosive tooth wear has become a growing public health problem [52]. It has been estimated that the worldwide prevalence of dental erosion in permanent teeth in children and adolescents is 30.4%, and in Europe 33.1% [53]. However, tooth erosion was found in less than 1% of Serbian schoolchildren in the present study. The reason might be that Serbian schoolchildren are less exposed to risk factors (acidic ingredients) than their peers in other countries. However, an explanation that is more possible is that the assessment protocol for diagnosis of tooth erosion used in the present study (WHO criteria) [6] is different from more precise tooth erosion criteria used in previous studies. Erosive tooth wear is a relatively new condition in pediatric dentistry, and it cannot be noticed easily, especially at the very beginning of its development. Uniform criteria for diagnosis of tooth erosion should be determined and used.

Traumatic tooth injuries are common during childhood. It was reported that 25% of all schoolchildren experience dental trauma [54]. Our data showed that the prevalence of dental injuries was 2% in both 12- and 15-year-olds. A higher prevalence (6.3–14%) of dental injuries in the same age groups was recently reported [30,55,56]. However, the finding from the present study should be considered with caution. Due to study design, only visible outcomes of previous tooth injuries (untreated or restored crown fractures) were recorded.

A high prevalence of malocclusions is a global oral health problem [57]. Malocclusions could pose functional, aesthetic, psychological, or social problems. Nationwide oral health surveys have the potential to provide information on dental needs of the population [58]. Quite a large number of studies on the prevalence of malocclusions in different populations were published. However, information on the prevalence of malocclusion in the Serbian population is scarce [5,59].

In the present study, oro-facial functions showed a slight improvement in 15-year-old children in comparison to 12-year-olds. Neutrocclusion was found in the majority of children, which is in accordance with global prevalence of Class I occlusion [60,61]. Findings of the previous epidemiological study in Serbia [5] also indicated high percentage of children with neutrocclusion (60.3% and 58.5% of 12-year-old and 15-year-old children, respectively). Similar results were presented in a national pathfinder survey in Greece [62]. Authors reported an Angle Class I molar relationship in 49.4% and 52.3% of 12-year-old and 15-year-old children, respectively. The difference in prevalence of sagittal malocclusions between neighboring countries may be attributed to ethnic differences and sample composition. Crowding, deep bite, and posterior crossbite were dominant malocclusion traits found in the present study, which is in accordance with the findings in the Greek survey [62]. Crowding was the most frequently reported malocclusion in Serbian children, as well as globally [61]. The prevalence of posterior crossbite in Serbia was slightly higher than in European countries, but within the global range [61]. Greek children showed similar prevalence of posterior crossbite with almost identical percentages in both age groups [62]. The prevalence of deep bite was higher in Serbian children in comparison to their peers in Europe [60]. Anterior crossbite and open bite had the lowest prevalence of all malocclusion traits, which is consistent with worldwide findings [60]. The lower prevalence of malocclusion in the present study, in comparison to the other parts of the world [61,63], could be either due to the composition of the sample or the growing trend of early initiation of orthodontic prevention and treatment in Serbia in recent years. The orthodontic findings from the present study could assist in providing orthodontic status from this part of the world, as well as in determining steps required for further improvement of measures in regards to preventive and treatment approaches.

The strengths of the present study should be highlighted. At the population level, this was the first epidemiological survey in the last fifteen years that evaluated oral health in Serbian children based on WHO methodology. Prevalence and severity of caries lesions were assessed using not only commonly used DMFT/DMFS indices to characterize cavitated caries lesions, but also recognizing initial enamel caries lesions, which increased the validity of the caries profile. The study provided valuable age- and area-specific indicators of oral health, which could give a foundation for future implementation of targeted preventive interventions.

There are a few limitations of the present study. Only healthy children were recruited for the survey, while children with disabilities were not examined. The organization and administration of community oral health services in Serbia differ among regions, thus influencing the available preventive measures and treatment modalities. Despite the limitations, the present study provides valuable oral health profile of Serbian schoolchildren, which can be used for updating and refining current preventive oral health care program.

## 5. Conclusions

Dental caries is still a common public health problem among schoolchildren in Serbia. The level of caries experience in 12-year-olds is low. However, a significant increase in prevalence of caries disease between 12- and 15-year-old groups was found. Given the specific needs and health-compromising behaviors of adolescents, prevention efforts need to be tailored very diligently. Improvement of oral hygiene habits is a priority to get better dental and gingival health. The existing school-based prevention program for oral health promotion needs additional attention. Corrective actions and reforms to the current oral health prevention program in Serbia are necessary to deliver strategies for addressing the specific challenges in schoolchildren.

## Figures and Tables

**Figure 1 ijerph-19-12269-f001:**
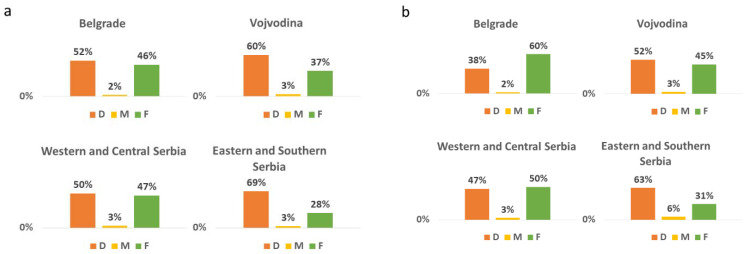
The distribution of DMFT according to the region in (**a**) 12-year-olds and (**b**) 15-year-olds in Serbia. Significant differences in the distribution of D and F components were found between the regions (*p* < 0.001, Kruskal–Wallis test).

**Figure 2 ijerph-19-12269-f002:**
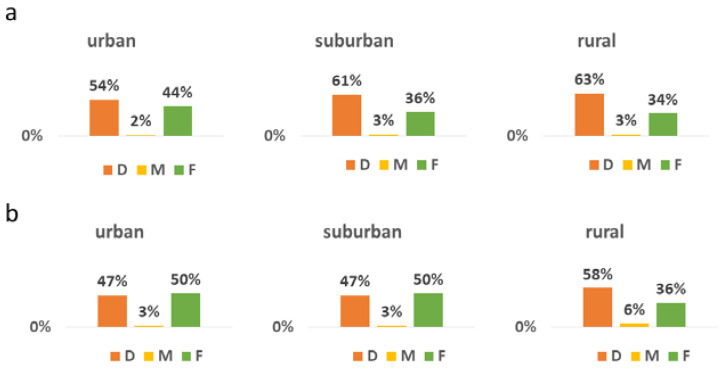
The distribution of DMFT according to the area in (**a**) 12-year-olds and (**b**) 15-year-olds in Serbia.

**Table 1 ijerph-19-12269-t001:** Results of intra-examiner and inter-examiner reproducibility.

		No. of Healthy Teeth	D	M	F	DMFT	Gingival Bleeding	PI
LEVEL 1								
Principal investigator 1–3	median	21	4	0	3	7	0.96	1.25
	min	20	4	0	3	7	0.93	1.21
	max	21	5	0	3	8	1.00	1.25
LEVEL 2								
Principal investigator 1 with examiners 4–14	median	14	14	0	0	14	0.64	1.39
	min	13	14	0	0	14	0.57	1.21
	max	14	15	0	0	15	0.71	1.46
Principal investigator 2 with examiners 15–25	median	17.5	9.5	0	1	10.5	0.96	1.93
	min	17	8	0	1	9	0.93	1.86
	max	19	10	0	1	11	0.96	2.00
Principal investigator 3 with examiners 26–36	median	17	9	0	2	11	0.64	1.18
	min	17	9	0	2	11	0.43	1.14
	max	17	9	0	2	11	0.64	1.29

**Table 2 ijerph-19-12269-t002:** Agreement level after examination of permanent teeth (Kappa (Kappa range between examiners)).

	Overall	Dental Status	Gingival Bleeding	PI
LEVEL 1	0.88 (0.84–0.93)	0.74 (0.63–0.91)	No computation	1.00
LEVEL 2	0.91 (0.89–0.96)	0.94 (0.91–1.00)	0.82 (0.70–0.91)	0.79 (0.76–0.96)

**Table 3 ijerph-19-12269-t003:** Prevalence of caries-free and children with severe caries (%), DMFT and DMFS values (mean ± SD (median)) in 12-year-old children in Serbia.

	No. of Participants	Caries-Free	Severe Caries	DMFT	DMFS
Gender					
Boys	613	41% ^a^	45%	2.14 ± 2.68 (1.00) ^A^	3.00 ± 4.58 (1.00) ^I^
Girls	587	31% ^a^	43%	2.51 ± 2.69 (2.00) ^A^	3.34 ± 4.37 (2.00) ^I^
Area					
Urban	603	41% ^b^	39% ^h^	2.01 ± 2.50 (1.00) ^B^	2.58 ± 3.65 (1.00) ^J^
Suburban	247	23% ^b,c^	55% ^h^	3.06 ± 2.81 (3.00) ^B,C^	4.47 ± 5.04 (3.00) ^J,K^
Rural	350	36% ^c^	45%	2.39 ± 2.39 (2.00) ^C^	3.42 ± 5.45 (1.00) ^K^
Administrative region					
City of Belgrade	250	50% ^d,e,f^	36% ^i^	1.64 ± 2.35 (0.50) ^D,E,F^	2.24 ± 4.04 (1.00) ^L,M,N^
Vojvodina	300	35% ^e^	46%	2.36 ± 2.74 (2.00) ^D^	3.17 ± 5.01 (1.00) ^L,O^
Central and Western Serbia	338	36% ^d^	37% ^j^	2.26 ± 2.61 (2.00) ^E^	3.06 ± 3.94 (2.00) ^M^
Southern and Eastern Serbia	312	28% ^f^	52% ^i,j^	2.62 ± 2.86 (2.00) ^F^	3.68 ± 4.42 ^N,O^
Parental employment					
Both parents employed	938	36% ^g^	42%	2.32 ± 2.67 (2.00) ^G^	3.17 ± 4.35 (2.00) ^P^
One parent employed	225	25% ^g^	51%	2.85 ± 2.97 (2.00) ^G,H^	3.99 ± 4.62 (2.00) ^P,Q^
Both parents unemployed	37	40%	28%	1.37 ± 1.54 (1.00) ^H^	2.43 ± 3.84 (1.00) ^Q^

^a, b, d, e, f, h^ *p* < 0.001, chi-square test, between the values marked with the same letter; ^c, g, i, j^ *p* < 0.05, chi-square test, between the values marked with the same letter; ^A^ *p* < 0.001, Mann–Whitney test, between the values marked with the same letter; ^B, C, G, H, J, K, L, O, P, Q^ *p* < 0.05, Kruskal–Wallis test, between the values marked with the same letter; ^D, E, F, M, N^
*p* < 0.001, Kruskal–Wallis test, between the values marked with the same letter; ^I^ *p* < 0.05, Mann–Whitney test, between the values marked with the same letter.

**Table 4 ijerph-19-12269-t004:** Prevalence of caries-free and children with severe caries (%), DMFT and DMFS values (mean ± SD (median)) in 15-years old children in Serbia.

	No. of Participants	Caries-Free	Severe Caries	DMFT	DMFS
Gender					
Boys	611	23%	63%	3.95 ± 3.80 (3.00)	5.79 ± 6.39 (4.00)
Girls	609	20%	59%	4.24 ± 3.81 (4.00)	5.87 ± 6.46 (4.00)
Area					
Urban	600	29% ^a,b^	54% ^h,i^	3.41 ± 3.70 (2.00) ^A,B^	4.60 ± 5.59 (3.00) ^H,I^
Suburban	263	15% ^a^	65% ^h^	4.62 ± 3.58 (4.00) ^A^	6.36 ± 5.96 (5.00) ^H^
Rural	357	17% ^b^	65% ^i^	4.58 ± 4.01 (4.00) ^B^	7.10 ± 7.58 (5.00) ^I^
Administrative region					
City of Belgrade	250	38% ^c,d,e^	53% ^j^	2.95 ± 3.48 (2.00) ^C,D,E^	3.73 ± 4.79 (2.00) ^J,K,L^
Vojvodina	312	19% ^c^	60%	4.35 ± 4.03 (4.00) ^C^	6.21 ± 6.87 (4.00) ^J^
Central and Western Serbia	358	18% ^d^	62%	4.39 ± 3.73 (4.00) ^D^	6.21 ± 6.25 (4.00) ^K^
Southern and Eastern Serbia	300	19% ^e^	66% ^j^	4.05 ± 3.68 (3.00) ^E^	6.23 ± 6.88 (4.00) ^L^
Parental employment					
Both parents employed	869	24% ^f^	58% ^k^	3.80 ± 3.67 (3.00) ^F^	5.25 ± 5.92 (4.00) ^M^
One parent employed	287	14% ^f,g^	67% ^k^	4.67 ± 3.78 (4.00) ^F,G^	7.35 ± 7.25 (6.00) ^M,N^
Both parents unemployed	64	28% ^g^	57%	3.88 ± 4.43 (3.00) ^G^	5.67 ± 7.39 (3.50) ^N^

^a, b, c, d, e^ *p* < 0.001, chi-square test, between the values marked with the same letter; ^f, g, h, i, j, k^ *p* < 0.05, chi-square test, between the values marked with the same letter; ^A, B, C, D, E, F, H, I, J, K, L, M^ *p* < 0.001, Kruskal–Wallis test, between the values marked with the same letter; ^G, N^ *p* < 0.05, Kruskal–Wallis test, between the values marked with the same letter.

**Table 5 ijerph-19-12269-t005:** Gingival health and oral hygiene.

	12-Year-Olds			15-Year-Olds		
	Gingivitis (%)	PI		Gingivitis (%)	PI	
		PI = 0 (%)	Mean ± SD (Median)		PI = 0 (%)	Mean ± SD (Median)
Gender						
Boys	27%	35%	0.73 ± 0.78 (0.50)	19%	32% ^E^	0.69 ± 0.72 (0.50) ^H^
Girls	24%	38%	0.69 ± 0.78 (0.38)	16%	41% ^E^	0.54 ± 0.70 (0.29) ^H^
Area						
Urban	26%	40% ^d^	0.68 ± 0.78 (0.33) ^k^	20% ^A^	36%	0.63 ± 0.71 (0.36)
Suburban	24%	36%	0.68 ± 0.74 (0.43)	8% ^A,B^	40%	0.53 ± 0.67 (0.29) ^I^
Rural	28%	28% ^d^	0.81 ± 0.83 (0.57) ^k^	22% ^B^	35%	0.67 ± 0.76 (0.43) ^I^
Administrative region						
City of Belgrade	29% ^a^	50% ^e,f^	0.50 ± 0.69 (0.06) ^l,m,n^	27% ^C,D^	44% ^F^	0.47 ± 0.63 (0.21) ^J,K^
Vojvodina	31%^c^	26% ^e,g,h^	0.78 ± 0.82 (0.46) ^l,o^	17% ^D^	32% ^F^	0.68 ± 0.75 (0.43) ^J,L^
Central and Western Serbia	18% ^a,b,c^	41% ^f,g^	0.88 ± 0.87 (1.00) ^m,p^	13% ^C^	36%	0.71 ± 0.80 (0.43) ^K,M^
Southern and Eastern Serbia	24% ^b^	36% ^h^	0.61 ± 0.64 (0.50) ^n,o,p^	20%	38%	0.47 ± 0.54 (0.29) ^L,M^
Parental employment						
Both parents employed	24%	43% ^i,j^	0.66 ± 0.77 (0.36) ^q,r^	17%	37% ^G^	0.57 ± 0.70 (0.29) ^N^
One parent employed	25%	26% ^i^	0.96 ± 0.84 (1.00) ^q^	21%	29% ^G^	0.72 ± 0.71 (0.57) ^N^
Both parents unemployed	33%	20% ^j^	1.03 ± 0.71 (1.00) ^r^	24%	40%	0.59 ± 0.68 (0.41)

^a, b, h, j, D, E, F, G^ *p* < 0.05, chi-square test, between the values marked with the same letter; ^c, d, e, f, g, i, A, B, C^ *p* < 0.001, chi-square test, between the values marked with the same letter; ^H^ *p* < 0.001, Mann–Whitney test, between the values marked with the same letter; ^k, n, o, p, r, I, L, M, N^ *p* < 0.05, Kruskal–Wallis test, between the values marked with the same letter; ^l, m, q, J, K^ *p* < 0.001, Kruskal–Wallis test, between the values marked with the same letter.

**Table 6 ijerph-19-12269-t006:** Orthodontic findings.

	12-Year-Olds	15-Year-Olds
Class I molar relationship	87%	90%
Overjet > 4 mm	7%	5%
Anterior crossbite	4%	3%
Deep bite	20%	19%
Anterior open bite	1%	2%
Posterior crossbite	10%	9%
Spacing	7%	6%
Crowding	25%	28%
Mouth breathing	6%	4%
Tongue thrust and dysfunctions	8%	5%

## Data Availability

Datasets generated and/or analyzed during the study are available from the first author on reasonable request.

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
