# Peer review of "Oral Health in 12- and 15-Year-Old Children in Serbia: A National Pathfinder Study"

_ijerph, 2022, doi:10.3390/ijerph191912269_

Round 1
Reviewer 1 Report
The study aimed to assess the oral status of 12-year-old and 15-year-old school-children in Serbia. The paper is interesting and adequately written; however, some corrections are necessary.
I ask the authors to correct the following:
The conclusion in the abstract is too general. Please base it on the results obtained.
Table 3 does not include the median for the factors DMFT and DMFS for Southern and Eastern Serbia. Please correct the same.
Figures 1 and 2 are illegible due to their size; please present the same data in a table to make them more visible or improve the picture resolution.
Please put Table 5 to one side, making it easier to follow.
The discussion is more about stating the results than comparing the results with studies conducted in Serbia or some other close countries in the region and the world especially Europe. Therefore, please make a better comparison of the results.
Please also list the strengths and weaknesses of the study.
It would be desirable if, in addition to employment, the professional education and financial status of the parents were taken into account, to see how certain factors such as dental caries and similar correlate with this.
Reviewer 2 Report
Dear authors, you have made a great article based on a huge study all over Serbia country, congratulations.
You have written a great article, the statistical part is correct, the population chosen and the statistical methods are correctly used. It is true that the comparison in the discussion are with articles of Serbia and I would suggest to try to compare with other countries and with more recent articles: https://doi.org/10.1016/j.heliyon.2022.e09557 For exampleThank you
Reviewer 3 Report
Journal: International Journal of Environmental Research and Public Health
(ISSN 1660-4601)
Manuscript ID: ijerph-1855263
Type of manuscript: Article
Title: Oral health in 12- and 15-year old children in Serbia. A national pathfinder study
This manuscript was intended to present the results of oral health surveys performed among 12-year and 15-year old children in Serbia according to the WHO methodology.
I was glad to read this concise and well written article which in my opinion fully comply with the requirements of that kind of report.
Only one remark:
All abbreviations obligatory should be defined at the first usage in the text including the abstract even though these abbreviations are well-known like ‘DMFT/DMFS’.
Round 2
Reviewer 1 Report
Thanks to the authors for accepting only some of the corrections. I still believe that the discussion is uninteresting and looks more like an introduction than an explanation and comparison of the results. Please remove the first four paragraphs from the discussion or move them to the introduction and explain the results in more detail compared with similar studies from the country or the world.
